# Enhanced Energy Storage Performance and Efficiency in Bi_0.5_(Na_0.8_K_0.2_)_0.5_TiO_3_-Bi_0.2_Sr_0.7_TiO_3_ Relaxor Ferroelectric Ceramics via Domain Engineering

**DOI:** 10.3390/ma16144912

**Published:** 2023-07-09

**Authors:** Srinivas Pattipaka, Hyunsu Choi, Yeseul Lim, Kwi-Il Park, Kyeongwoon Chung, Geon-Tae Hwang

**Affiliations:** 1Department of Materials Science and Engineering, Pukyong National University, 45 Yongso-ro, Nam-Gu, Busan 48513, Republic of Korea; cnuphy444@gmail.com (S.P.); sky5021184@pukyong.ac.kr (H.C.);; 2School of Materials Science and Engineering, Kyungpook National University, 80 Daehak-ro, Buk-Gu, Daegu 41566, Republic of Korea; kipark@knu.ac.kr; 3Department of Biofibers and Biomaterials Science, Kyungpook National University, Daegu 41566, Republic of Korea; kychung@knu.ac.kr

**Keywords:** lead-free ceramic capacitors, dielectric, relaxor ferroelectric, domain engineering, energy storage

## Abstract

Dielectric materials are highly desired for pulsed power capacitors due to their ultra-fast charge-discharge rate and excellent fatigue behavior. Nevertheless, the low energy storage density caused by the low breakdown strength has been the main challenge for practical applications. Herein, we report the electric energy storage properties of (1 − *x*) Bi_0.5_(Na_0.8_K_0.2_)_0.5_TiO_3_-*x*Bi_0.2_Sr_0.7_TiO_3_ (BNKT-BST; *x* = 0.15–0.50) relaxor ferroelectric ceramics that are enhanced via a domain engineering method. A rhombohedral-tetragonal phase, the formation of highly dynamic PNRs, and a dense microstructure are confirmed from XRD, Raman vibrational spectra, and microscopic investigations. The relative dielectric permittivity (2664 at 1 kHz) and loss factor (0.058) were gradually improved with BST (*x* = 0.45). The incorporation of BST into BNKT can disturb the long-range ferroelectric order, lowering the dielectric maximum temperature *T_m_* and inducing the formation of highly dynamic polar nano-regions. In addition, the *T_m_* shifts toward a high temperature with frequency and a diffuse phase transition, indicating relaxor ferroelectric characteristics of BNKT-BST ceramics, which is confirmed by the modified Curie-Weiss law. The rhombohedral-tetragonal phase, fine grain size, and lowered *T_m_* with relaxor properties synergistically contribute to a high *P_max_* and low *P_r_*, improving the breakdown strength with BST and resulting in a high recoverable energy density *W_rec_* of 0.81 J/cm^3^ and a high energy efficiency *η* of 86.95% at 90 kV/cm for *x* = 0.45.

## 1. Introduction

Materials with high energy and power have received extensive attention for high-power applications, such as microwaves, electromagnetic devices, pulsed power devices, hybrid electric vehicles, high-frequency inverters, and other energy storage devices [1,2,3]. In particular, dielectric ceramics are the most promising materials for energy storage applications due to their super-fast charge-discharge rate and excellent temperature stability compared to electrochemical energy storage devices (batteries and electrochemical capacitors) and dielectric polymers [4,5,6]. However, the dielectric capacitor’s energy storage density and efficiency are much lower than those of polymers/batteries due to their low dielectric breakdown strength (DBS), which restricts their practical application in energy storage devices.

The recoverable energy density (*W_rec_*) of a dielectric capacitor is governed by the applied electric field (*E*) and induced polarization (*P*), expressed by the following equation, usually estimated from the *P-E* loop, and it is schematically shown in Figure 1a by the shaded area with cyan color [7,8,9,10].
(1)$$W_{rec}=\int_{P_{r}}^{P_{max}}E\;dP$$
where *P_max_* and *P_r_* are the maximum polarization and remnant polarization, respectively (Figure 1a). Energy efficiency (*η*) can be estimated by the following equation [8,9,10].
(2)$$\eta =\frac{W_{rec}}{W_{rec}+W_{loss}}$$
where *W_loss_* is the hysteresis loss. According to Equation (1), the energy storage properties can be significantly enhanced by increasing the difference between *P_r_* and *P_max_* (Δ*P*). The breakdown electric field (*E_BD_*) is also an essential factor for energy storage; i.e., a higher DBS is responsible for a large energy storage density.

Linear dielectrics (LDEs), ferroelectrics (FEs), relaxor ferroelectrics (RFEs), and anti-ferroelectrics (AFEs) have been widely explored for electrostatic energy storage applications [11]. LDEs (Al_2_O_3_ and SrTiO_3_) typically show low relative dielectric permittivity and loss factor, high *E_BD_*, and free hysteresis loop with low polarization, resulting in poor *W_rec_* and high *η* [11,12,13]. The AFEs (PbZrO_3_, PbHfO_3_, AgNbO_3_, and NaNbO_3_) display elevated polarization and substantial hysteresis because of phase transition between AFE and FE phases induced by the external field, leading to extremely high *W_rec_* and low *η* [14,15,16,17,18]. The RFEs (Bi_0.5_Na_0.5_TiO_3_ (BNT), BaTiO_3_ (BT), and BiFeO_3_ (BFO)) display moderate *P_max_* and small *P_r_* (Figure 1b), arising from widespread PNRs, which generally exhibit highly dynamic short-range FE orders. However, these PNRs gradually transform into long-range FE orders with an increasing field, resulting in large *P_max_*. After eliminating the electric field, the induced FE orders will easily revert to PNRs, leading to small *P_r_* [19,20,21,22]. Hence, the RFEs usually show high *W_rec_* and *η*. Therefore, lead-based RFE materials have been widely investigated for energy storage applications [23,24]. However, lead is hazardous to the environment and human health due to its toxicity, which has motivated the development of alternative lead-free materials. In recent years, lead-free perovskite-structured (ABO_3_) RFEs, such as BT [25,26,27], BNT [6,28,29,30,31], BFO [32,33,34], and other lead-free perovskite RFEs, such as Bi_4_Ti_3_O_12_ [35], Sr_1.25_Bi_2.75_Nb_1.25_Ti_1.75_O_12_ [36], and Sr_0.6_Ba_0.4_Nb_2_O_6_ [37] based materials with boosted energy storage performance, have been reported for applications in energy storage devices.

Perovskite-structured BNT-based ceramics exhibit a strong ferroelectric response, since Bi^3+^ has a lone pair of electrons (6s^2^), which strongly hybridizes with the oxygen 2p orbital [38]. Furthermore, the formation of highly dynamic polar nano-regions (PNRs) are facilitated by local random fields induced by valency differences and compositional inhomogeneity [39,40]. Moreover, the relaxor behavior of the material can be improved by adding another phase with a similar perovskite to form a solid solution or modifying the base compound with a suitable dopant, which enables slim *P-E* loops [41]. Sayyed et al. [42] investigated the local structural deformation and dielectric anomalies near the morphotropic phase boundary (MPB) of (1 − *x*) Na_0.5_Bi_0.5_TiO_3_-*x*SrTiO_3_ ceramics. The ferroelectric response of (1 − *x*) Na_0.5_Bi_0.5_TiO_3_-SrTiO_3_-*x*AgNbO_3_ ceramics is similar to the antiferroelectric response and improved energy storage performance [43]. Shi et al. [44] reported that Zr- and Sm-doped 0.74Na_0.5_Bi_0.5_TiO_3_-0.26SrTiO_3_ ceramics significantly enhanced energy storage performance and the DBS. Bi_0.2_Sr_0.7_TiO_3_ (BST) exhibits strong polarization and a wide-phase transition temperature with diffused dielectric maxima. It was incorporated into BNT ceramics, suppressing the field-generated ferroelectric phase and achieving a large *P_max_* and small *P_r_* [45,46]. Recently, Li et al. reported a synergistic approach to enhance the energy storage response in BNT-based RFEs by introducing PNRs and lowering the transition temperature by stabilizing the AFE responses at low temperatures [9].

In this work, we investigate a domain engineering process to improve the energy storage performance by modifying Bi_0.5_(Na_0.8_K_0.2_)_0.5_TiO_3_ (BNKT) RFEs with BST, since Bi_0.5_(Na_1−*x*_K*_x_*)_0.5_TiO_3_ exhibits a stronger ferroelectric response with relaxor behavior at the MPB at *x* = 0.16–0.2 [47,48] than pure BNT. It is revealed that the addition of BST can disturb the long-range ferroelectric order and transform the ferroelectric microdomains of BNKT into highly dynamic PNRs. This results in a macroscopic ferroelectric to relaxor ferroelectric transition, as schematically illustrated in Figure 1b. The favorable relaxor ferroelectric state formed by the domain engineering method simultaneously produces a large *P_max_* and reduced *P_r_*, which facilitates the enhancement in DBS with the BST, resulting in high energy density and high efficiency of the BNKT-BST RFEs.

## 2. Materials and Methods

(1 − *x*) Bi_0.5_(Na_0.8_K_0.2_)_0.5_TiO_3–*x*_Bi_0.2_Sr_0.7_TiO_3_ (BNKT-BST; *x* = 0.15, 0.30, 0.40, 0.45, and 0.50) RFE ceramics were fabricated via a conventional solid-state reaction method. To prepare BNKT and BST, high-purity raw materials of Bi_2_O_3_ (Sigma-Aldrich, St. Louis, MI, USA, 99.9%), Na_2_CO_3_ (Sigma-Aldrich, St. Louis, MI, USA, 99.5%), K_2_CO_3_ (Sigma-Aldrich, St. Louis, MI, USA, 99%), TiO_2_, (Sigma-Aldrich, St. Louis, MI, USA, 99%), and SrCO_3_ (Sigma-Aldrich, St. Louis, MI, USA, 98%) were weighed according to the nominal stoichiometric compositions and then ball-milled using a planetary ball mill for 24 h with ZrO_2_ balls in ethanol. After the slurries were dried at 120 °C, the mixture of BNKT and BST powders was calcined at 800 °C and 950 °C for 2 h and 3 h, respectively, to form a pure phase of Bi_0.5_(Na_0.8_K_0.2_)_0.5_TiO_3_ and Bi_0.2_Sr_0.7_TiO_3_. Both BNKT and BST calcined powders were mixed and ball-milled for 12h to prepare a BNKT-BST composition. Further, these powders were granulated with 5 wt.% polyvinyl alcohol (Sigma-Aldrich, 99%, St. Louis, MI, USA,) and uniaxially pressed into disks, at a pressure of 10 MPa, of 10 mm diameter and ~0.5 mm thickness, followed by sintering at 1100 °C for 3 h. To perform electrical measurements, a silver paste (ELCOAT, Electroconductives) was coated on both sides of the sintered disks.

The phase formation of the BNKT-BST ceramic samples was examined using an X-ray diffractometer (Rigaku, Tokyo, Japan, TTRAX III 18 kW) with monochromatic Cu-Kα radiation (*λ* = 1.5406 Å). Raman spectra were recorded using a Raman spectrometer (JOBIN YVON, Oberursel, Germany, LABRAM HR800) with a laser wavelength of 532.06 nm. Surface morphology was investigated using a field emission scanning electron microscope (FESEM) (JEOL, Tokyo, Japan, JSM-7610F). Temperature- and frequency-dependent dielectric properties were measured from room temperature (RT) to 450 °C and 1 kHz–1 MHz using an impedance analyzer (Hewlett Packard, Palo Alto, CA, USA, 4294A). *P-E*, *I-E* loops, and fatigue behavior were measured using a ferroelectric tester (AixACCT Systems GmbH, Aachen, Germany, TF Analyzer 2000).

## 3. Results and Discussion

### 3.1. Phase Evolution and Microstructure

Figure 2a shows the X-ray diffraction (XRD) patterns of BNKT-BST ceramics (*x* = 0.15–0.50) in the 2θ range of 20–70°. All of the samples revealed a rhombohedral and tetragonal crystal structure, indicating the diffusion of BST into BNKT and the formation of BNKT-BST as a homogeneous solid solution. At RT, the BNKT system exhibits a rhombohedral and tetragonal crystal structure near MPB at *x* = 0.16–0.2 [47,48]. The formation of MPB in BNKT-BST ceramics is confirmed by the splitting of the (021)/(111) and (122)/(211) peaks at 2θ around 40° and 58°, respectively, which is shown in Figure 2b. Similar splitting and formation of MPB were observed in Bi_0.5_(Na_1–*x*_K*_x_*)_0.5_TiO_3_]-BiAlO_3_ [49], Bi_0.5_Na_0.5_TiO_3_-Bi_0.5_K_0.5_TiO_3_-Bi_0.5_Li_0.5_TiO_3_ [50], and (Bi_0.5_Na_0.5_)TiO_3_-(Bi_0.5_K_0.5_)TiO_3_-BaTiO_3_ [51] ceramics. In addition, both the (021) and (122) peaks shifted slightly toward lower angles with increasing BST into BNKT, demonstrating enhanced lattice parameters (Figure 2b). The enhancement in lattice parameters can be attributed to the ionic radius of Sr^2+^ (1.44 Å), which is larger than that of Bi^3+^ (1.36 Å), Na^+^ (1.39 Å), and K^+^ (1.38 Å), respectively, at the *A*-site [52,53,54].

Figure 2c shows the Raman spectra of BNKT-BST along with spectral de-convolution in the Raman shift of 50–1000 cm^−1^. The Raman spectra of all compositions are similar to the previous reports of BNKT-based ceramics [51,55]. The Raman active bands are divided into four Raman vibration modes, as shown at the top of Figure 2c. (i) The modes below 200 cm^−1^ are related to the vibration of the *A*-site (Bi-O, Na-O, K-O, and Sr-O); (ii) the modes between 200 and 440 cm^−1^ correspond to the vibrations of B-O (Ti-O); (iii) the modes between 440 and 700 cm^−1^ correspond to the vibrations of BO_6_ (TiO_6_)-octahedra; and (iv) the modes above 700 cm^−1^ are related to the *A1* and *E* (longitudinal optical) overlapping modes [55]. The modes appearing at 124–172 cm^−1^ and 768 cm^−1^ are shifted to the higher wavenumbers of 128–189 cm^−1^ and 779 cm^−1^ with BST, associated with the *A*-site and *A1* + *E* vibrations caused by *A*-site disorder. Such a disorder is induced by the incorporation of BST (Bi^3+^ and Sr^2+^) into the BNKT (Bi^3+^, Na^+^, and K^+^) system [56]. In addition, a noticeable change at 250 and 320 cm^−1^ shifted toward a lower wavenumber of 233 and 305 cm^−1^ with BST, which is caused by an increase in the *B*-site disorder in the BNKT-BST system [53]. Moreover, these modes are slightly broadened, confirming the disturbance of the long-range ferroelectric order and the formation of highly dynamic PNRs, improving the relaxor characteristics of BNKT-BST [44]. This result is consistent with the XRD and electrical properties presented in Section 3.2 and Section 3.3.

Figure 3 shows FESEM images of the BNKT-BST ceramics. All of the ceramics display rectangular-shaped grains, which are homogeneously distributed. Figure 3d clearly shows that the *x* = 0.45 composition exhibits a highly dense microstructure and is more compact with smaller grains, as compared to the other samples of BNKT-BST (*x* < 0.45 and *x* = 0.50). To prove that all of the samples are homogeneous and highly dense, the density of sintered BNKT-BST ceramic samples was calculated using the Archimedes principle. It increased with BST from 5.58 g/cm^3^ to 5.73 g/cm^3^ for *x* = 0.15 to 0.45 and further decreased (5.54 g/cm^3^) for *x* = 0.50. The calculated relative density of BNKT-BST ranged from 94.96% to 98.17% of the theoretical density [57], confirming that these samples are homogeneous and highly dense. Further, the average grain size of the BNKT-BST ceramics was estimated using Image-J software (Wayne Rasband and contributors, National Institutes of Health, USA, ImageJ 1.53t) via the linear intercept method and found to be 1.37 µm for the *x* = 0.15 composition; it gradually enhanced to 1.6 µm with the incorporation of BST. Grain size enhancement is caused by the generation of oxygen vacancies by Sr^2+^ entering the perovskite of BNKT and being substituted at the *A*-site of Bi^3+^, Na^+^, and K^+^ [58]. Previous reports have investigated that the fine grain size with a homogeneous and dense microstructure can withstand higher electric fields, leading to high DBS, and improve energy storage performance [59,60].

### 3.2. Dielectric Properties and Relaxor Behavior

Figure 4a displays the frequency variation in the relative dielectric permittivity (*ε_r_*) (solid line) and loss factor (tan *δ*) (dot line) of BNKT-BST ceramic capacitors, measured at RT in the range of 1 kHz to 1 MHz. The sample *x* = 0.15 displayed a higher *ε_r_* of 1481 and tan *δ* of 0.231 at 1 kHz than pure BNKT (*ε_r_* of 1273 and tan *δ* of 0.047 at 1 kHz), as reported in our previous report [48]. These *ε_r_* values gradually enhanced to 2664, and the tan *δ* values reduced to 0.058 for the *x* = 0.45 sample (Figure 4b). The enhancement in the dielectric properties is attributed to the incorporation of BST into BNKT and the dense microstructure.

Figure 4c,d displays the temperature dependence of *ε_r_* and tan *δ* of BNKT-BST for the lower and higher compositions (*x* = 0.15 and 0.45), measured at various frequencies (0.1 kHz to 1 MHz). It was observed that the dielectric maximum temperature (*T_m_*) shifted toward a lower temperature 53 °C with the incorporation of BST for *x* = 0.45 (Figure 4d), as compared to *x* = 0.15 (*T_m_
*= 345 °C) (Figure 4c) and pure BNKT (300 °C) [48], and this is similar to the 0.74 Na_0.5_Bi_0.5_TiO_3_-0.26 SrTiO_3_ ceramics reported by Shi et al. [44]. The incorporation of BST into BNKT can disturb the long-range ferroelectric order, resulting in a lowered *T_m_*. This lower *T_m_* leads to the formation of highly dynamic PNRs due to the mismatch of the ionic radius at the *A*-site of BNKT-BST. In addition, the *T_m_* shifted toward higher temperatures, and dielectric peaks diffused with an increase in frequency. This frequency dispersion with a diffuse phase transition reveals typical relaxor ferroelectric characteristics [61,62]. The degree of the relaxor characteristics was determined using the modified Curie-Weiss law via the following equation [52,63].
(3)1εr−1εrm=T−TmγC
where $\varepsilon_{r}^{m}$ is the maximum relative dielectric permittivity at the maximum temperature *T_m_*, *T* is the temperature, *γ* is the degree of relaxation, and *C* is the Curie constant. Generally, the *γ* value is 1 for normal ferroelectrics and between 1 and 2 for relaxor ferroelectrics [63]. The insets of Figure 4c,d show the log-log plots of (1/$\varepsilon_{r}\;-\;$1/$\varepsilon_{r}^{m}$) vs. (*T* − *T_m_*) of BNKT-BST for *x* = 0.15 and 0.45, measured at 1 MHz. The value of *γ* slightly increased from 1.80 to 1.83, proving that there is an increase in relaxor behavior with BST from *x* = 0.15 to 0.45, leading to an increase in the energy storage performance. This is consistent with Raman’s results and previous reports [4,5,6].

### 3.3. FE-RFE Transformation, Domain Evolution, and Energy Storage Performance

Figure 5 displays the RT bipolar *P-E* hysteresis loops and current (*I*)-electric field (*E*) curves of BNKT-BST ceramic capacitors measured at various electric fields and 10 Hz. The BNKT-BST (*x* = 0.15) sample exhibits a typical ferroelectric (FE) characteristic, displaying high remnant polarization *P_r_* of 19.89 µC/cm^2^, high maximum polarization *P_max_* of 31.46 µC/cm^2^, and a high coercive field *E_c_* of 16.66 kV/cm. These values, listed in Table 1, gradually decreased, whereas the *E_max_* or *E_BD_* increased from 57.42 kV/cm to 90 kV/cm with the incorporation of BST (*x* = 0.45), which is favorable for high energy storage density (Figure 5f). It is evident that the two peaks in the *I-E* curves (*x* ≥ 0.30) and slim *P-E* loops are attributed to the formation of highly dynamic PNRs, which can commonly be seen in RFEs [64]. In general, the *P-E* loops present in normal FEs are due to the macroscopic domain wall motion, while in RFEs, highly dynamic PNRs exist instead of macrodomains, resulting in slim *P-E* loops [19].

Further, the *W_rec_* was calculated via Equation (1) from *P-E* loops, which are shown in Figure 6 (cyan shaded area). The *W_loss_* is calculated by the enclosed area of the *P-E* loops in the first quadrant (magenta shaded area), and *η* is calculated by Equation (2); they are listed in Table 1. The *W_rec_* values gradually increased, the *W_loss_* values decreased with the substitution of BST, and the composition *x* = 0.45 displays a high energy density of 0.81 J/cm^3^ at an *E_BD_* of 90 kV/cm and high energy efficiency of 86.95% (Figure 6f). The improvement in the energy storage performance is achieved via the domain engineering method by modifying BNKT with BST. It can be understood that the substitution of BST can transform the ferroelectric microdomains of BNKT into highly dynamic PNRs, resulting in a macroscopic FE to RFE transition. This domain evolution and transformation of FE to RFE transition in the present samples is schematically shown in Figure 1b. The highly dynamic PNRs induced large *P_max_* and low *P_r_*, which improved the DBS with the incorporation of BST, resulting in high energy storage density and high energy efficiency of the BNKT-BST RFEs [65]. The obtained *W_rec_* and *η* of 0.55 BNKT-0.45 BST are comparable/superior to other lead-free RFEs and are promising for energy storage capacitors [64,65,66,67,68,69].

Electrical fatigue endurance is an important property necessary for energy storage applications. Therefore, the fatigue behavior of BNKT-BST ceramic capacitors was measured up to 10^6^ electric cycles at a frequency of 10 Hz under an electric field of 90 kV/cm. Figure 7 shows the unipolar *P-E* loops of BNK-BST (*x* = 0.45) and corresponding *W_rec_* (square line) and *η* (circle line) values measured after various electric cycles (black, red and blue colour *P*-*E* loops measured at 10^0^ and 10^3^ and 10^6^, respectively, as shown in inset of Figure 7). It is observed that the slender *P-E* loops are without significant change, revealing an excellent fatigue-free response and negligible variations in *W_rec_* and *η*.

## 4. Conclusions

The domain-engineered relaxor ferroelectric BNKT-BST lead-free ceramics were fabricated by a solid-state reaction method and demonstrated structural, microstructural, dielectric, and ferroelectric properties in detail. XRD, Raman spectra, and FESEM studies revealed the formation of a rhombohedral-tetragonal phase, highly dynamic PNRs, and dense microstructure. The dielectric properties were improved with BST, and a high *ε_r_* of 2664 and low tan *δ* of 0.058 at 1 kHz were obtained for the *x* = 0.45 composition. The incorporation of BST into BNKT can disturb the long-range ferroelectric order, causing lowered *T_m_* and the formation of highly dynamic PNRs. In addition, the *T_m_* shifts toward a high temperature with frequency and diffuse phase transition, indicating relaxor ferroelectric characteristics of BNKT-BST ceramics, and is confirmed via the modified Curie-Weiss law. The rhombohedral-tetragonal phase, fine grain size, and lowered *T_m_* with relaxor properties simultaneously contribute to a high *P_max_* and low *P_r_*. This improves the DBS and gives rise to giant energy storage density and high energy efficiency of the BNKT-BST RFEs, making this material a good candidate for pulse-driving energy storage applications.

## Figures and Tables

**Figure 1 materials-16-04912-f001:**
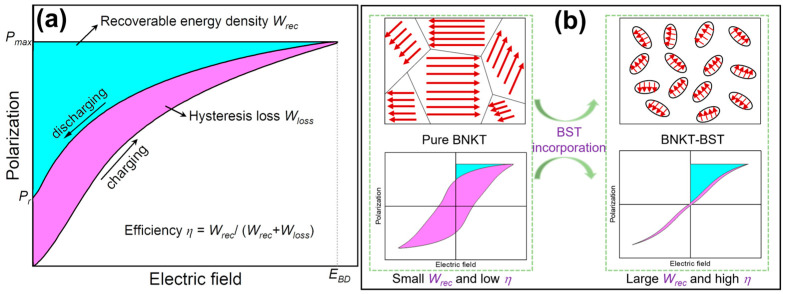
Schematic diagram of (**a**) recoverable energy density and hysteresis loss from *P-E* hysteresis loop of a dielectric material. (**b**) Domain evolution and formation of FE to RFE transition with the substitution of BST into BNKT, resulting in enhanced *W_rec_* and *η*.

**Figure 2 materials-16-04912-f002:**
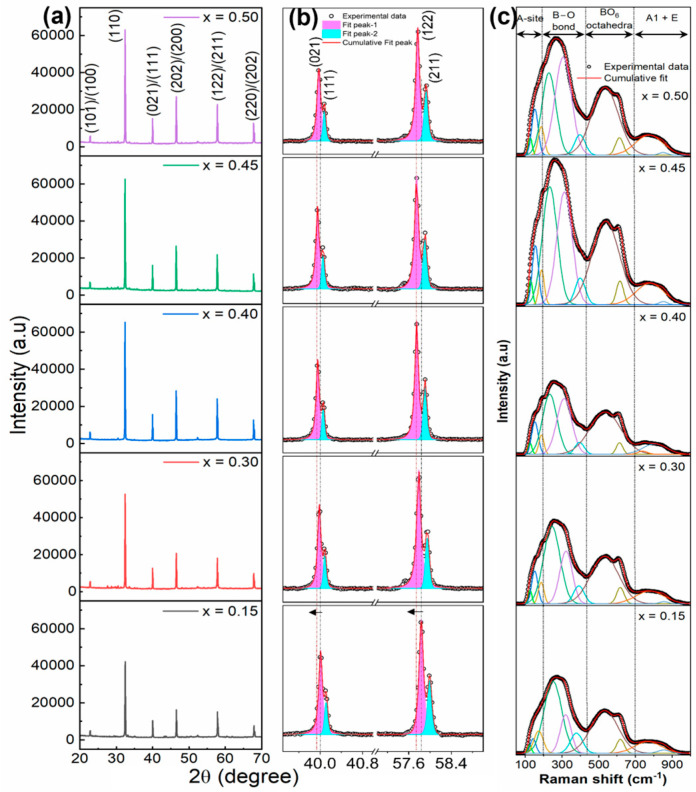
(**a**) XRD patterns of the (1 − *x*)BNKT-*x*BST ceramics for *x* = 0.15–0.50, where (**b**) 2θ = 39–59°. (**c**) Raman spectra of BNKT-BST ceramics along with spectral deconvolution.

**Figure 3 materials-16-04912-f003:**
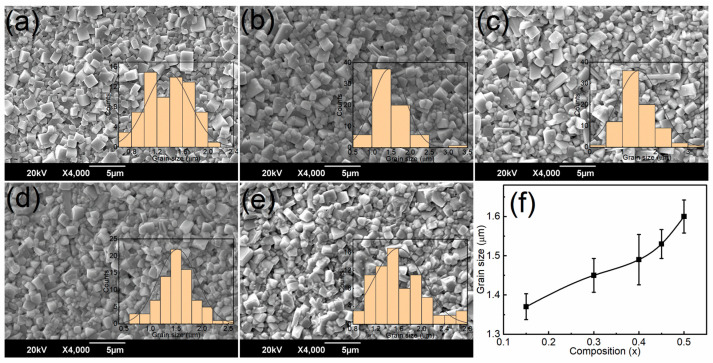
FESEM images of (1 − *x*) BNKT-*x*BST ceramics for (**a**) *x* = 0.15, (**b**) *x* = 0.30, (**c**) *x* = 0.40, (**d**) *x* = 0.45, and (**e**) *x* = 0.50. The inset of (**a**–**e**) shows the average grain size versus counts (grain size distribution histogram). (**f**) The variation in grain size with composition (*x*).

**Figure 4 materials-16-04912-f004:**
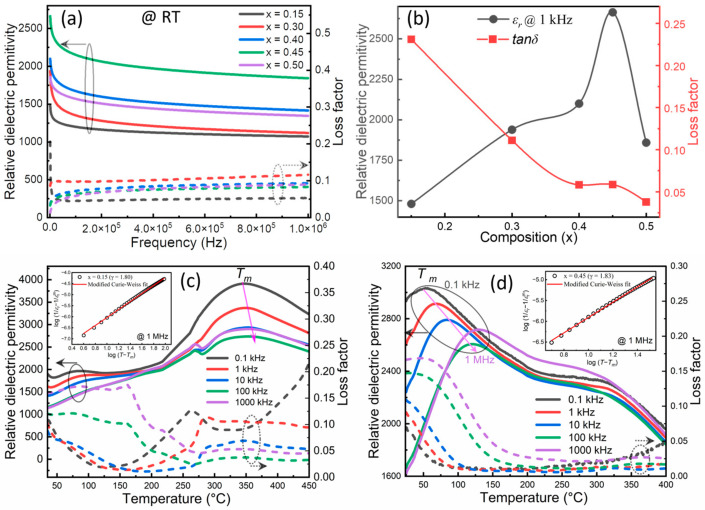
(**a**) Frequency variation of relative dielectric permittivity and loss factor of BNKT-BST ceramics for *x* = 0.15–0.50. (**b**) Composition vs. relative dielectric permittivity and loss factor. (**c**,**d**) Temperature variation of relative dielectric permittivity and loss factor of BNKT-BST for *x* = 0.15 and 0.45 (The left and right sides of the arrows with circles enclosed by curves indicate relative dielectric permittivity and loss factor, respectively). The inset of (**c**,**d**) shows the $log\left(T-T_{m}\right)$ versus log1εr−1εrm of BNKT-BST for *x* = 0.15 and 0.45, respectively, at 1 MHz.

**Figure 5 materials-16-04912-f005:**
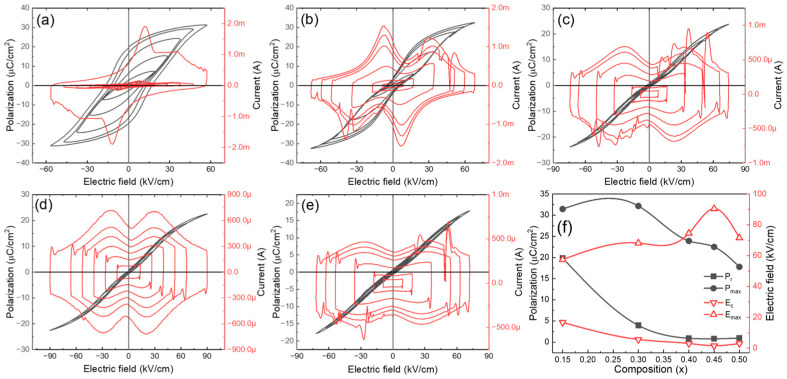
RT *P-E* and *I-E* curves of BNKT-BST ceramics for (**a**) *x* = 0.15, (**b**) *x* = 0.30, (**c**) *x* = 0.40, (**d**) *x* = 0.45, and (**e**) *x* = 0.50. (**f**) Composition (*x*) versus polarization and electric field.

**Figure 6 materials-16-04912-f006:**
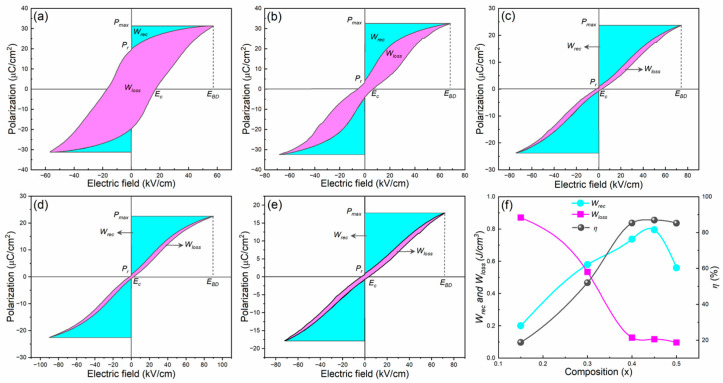
*P-E* loops of BNKT-BST ceramics measured at *E_BD_* and 10 Hz for (**a**) *x* = 0.15, (**b**) *x* = 0.30, (**c**) *x* = 0.40, (**d**) *x* = 0.45, and (**e**) *x* = 0.50. (**f**) Composition (*x*) versus *W_rec_*, *W_loss_*, and *η*.

**Figure 7 materials-16-04912-f007:**
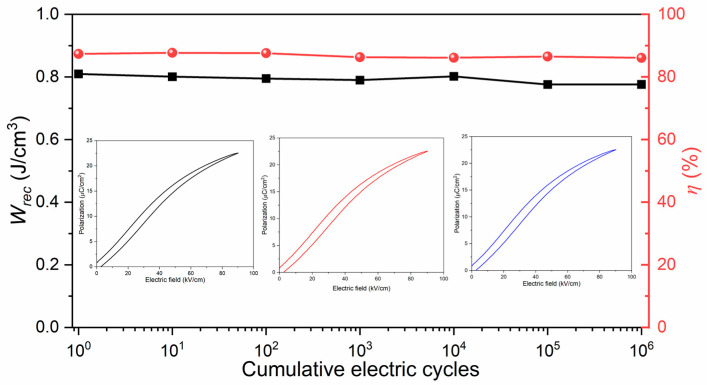
Fatigue behavior of BNKT-BST for *x* = 0.45 composition measured up to 10^6^ electric cycles.

**Table 1 materials-16-04912-t001:** List of all values of ferroelectric properties (*P_r_*, *P_max_*, *E_c_,* and *E_BD_*) and energy storage performance (*W_rec_* and *η*) of BNKT-BST ceramics for *x* = 0.15–0.50.

Composition	*P_r_* (µC/cm^2^)	*P_max_* (µC/cm^2^)	*E_c_* (kV/cm)	*E_BD_* (kV/cm)	*W_rec_* (J/cm^3^)	*η* (%)
*x* = 0.15	19.89	31.46	16.66	57.42	0.20	18.67
*x* = 0.30	3.95	32.18	5.59	68.18	0.57	52.02
*x* = 0.40	0.92	23.91	2.98	74.57	0.73	85.30
*x* = 0.45	0.78	22.5	1.58	90	0.81	86.95
*x* = 0.50	0.96	17.79	2.94	71.54	0.56	85.23

## Data Availability

Not applicable.

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
