# Peer review of "Enhanced Energy Storage Performance and Efficiency in Bi0.5(Na0.8K0.2)0.5TiO3-Bi0.2Sr0.7TiO3 Relaxor Ferroelectric Ceramics via Domain Engineering"

_materials, 2023, doi:10.3390/ma16144912_

Round 1
Reviewer 1 Report
In this work, authors report on the preparation of new ferroelectric materials. From the point of view of the novelty of the results obtained, the work is certainly relevant.
The phase composition and structure of the samples obtained are beyond doubt, since the authors used a combination of diffraction and vibrational methods for phase and structural certification, which makes it possible to remove most of the questions that arise when characterizing such low-symmetry samples.
Thermoelectric studies were carried out at a high technical level. The results are fully interpreted. The conclusions of the work correspond to the results obtained.
The article is written neatly and clearly. The reference list is representative. No inappropriate self-quoting.
In terms of the quality of the design of the graphic material, the work leaves a pleasant impression.
In general, in terms of the relevance and novelty of the results obtained, as well as the quality of the manuscript, I can recommend this work for publication.
Reviewer 2 Report
The Author presents the idea with as nothing related to the time. The paper should be improved by adding the time calculations for most experiments, especially for dielectric breakdown strength.
The author discussed the power storage Enhancement. However, the paper needs more work to be published. The article should have included the most crucial feature as follows.
1- The main question of the manuscript needs to be more precisely understandable.
2- The manuscript discussed energy improvement storge; this field was discussed extensively in many papers, and the author missed discussing the gap in this field or at least presented it as discussed in other literature
3- There is no comparison between the author’s and published results, which made the paper weak in presenting the results data.
4- The author should discuss the time for this improved storage all over the paper and justify it at each process level.
5- The conclusions are consistent but need to connect between the abstract (main question of the research) and results.
6- Published years 2023 and 2022 are very low numbers compared to other references.
7- Figure 4 and Figure 5 are both discussed with no reasons for the behavior; the author should add more effort to justify the behavior of the results.
8- Figure 3 x-axis and y-axis of the histogram must be clarified.
Reviewer 3 Report
This manuscript presents the synthesis of Bi0.5(Na0.8K0.2)0.5TiO3–Bi0.2Sr0.7TiO3 phases and discusses their detailed enhanced electrical properties and energy storage application. The samples are well characterized, and the data are explained well. However, some following issues should be addressed before accepting in Materials.
1. The author should add in the introduction the other lead-free perovskite-structured compounds with the relaxor ferroelectric behavior via substitution pathway that also exhibit potential energy storage performances. These articles can be cited: https://doi.org/10.1016/j.jssc.2023.124150 ; https://doi.org/10.1016/j.jallcom.2022.167809
2. Author mentioned that the XRD peaks shifted slightly towards lower angles, however from Figure 2b, it is difficult to see the shift, the author can add a guideline to see the shift as the x composition increases.
3. Author mentioned that the x=0.45 composition exhibits a highly dense microstructure (Fig 3d), it is possible to calculate the relative density values of each sample to prove that all samples are homogeneous and highly dense.
4. In the introduction, the authors stated that increased relaxor properties lead to increased energy storage performances. Therefore, the author should add the temperature dependence of dielectric permittivity and loss from the low x composition (at least 1 composition for comparison, better for all compositions) to prove that the relaxor behavior at x = 0.45 (the highest composition) is increased. The author can also compare the degree of relaxation γ.
5. I suggest the author list all values of ferroelectric properties (Pr, Pmax, Ec) and energy storage performance (energy density and efficiency) for all compositions in a Table; it is easier to compare all properties with increasing x composition. From the recent manuscript, the author only shows the values for the composition x=0.45 in the discussion, it would be better if all the values can be compared.
6. Some minor corrections in the manuscript:
- Adding the trademark of PVA and silver paste used in the materials section.
- The caption of Fig 4 should be revised. Temperature variation of dielectric should be (c) not (d).
- All symbols and abbreviations are better italicized (i.e., Pr, Pmax, Wloss, P-E, “A”-site, x, etc.) recheck all on the manuscript.
Round 2
Reviewer 2 Report
Accept in present form
Reviewer 3 Report
I am satisfied that the authors revise the manuscript as per the suggestions of the reviewers and explain the revision clearly in response letter, the present manuscript can be accepted for publication in Materials.